# Dynamic Maximum Entropy Reduction

**DOI:** 10.3390/e21070715

**Published:** 2019-07-22

**Authors:** Václav Klika, Michal Pavelka, Petr Vágner, Miroslav Grmela

**Affiliations:** 1Department of Mathematics—FNSPE, Czech Technical University, Trojanova 13, 12000 Prague, Czech Republic; 2Mathematical Institute, Faculty of Mathematics and Physics, Charles University, Sokolovská 83, 18675 Prague, Czech Republic; 3Weierstrass Institute, Mohrenstrasse 39, 10117 Berlin, Germany; 4École Polytechnique de Montréal, C.P.6079 suc. Centre-ville, Montréal, QC H3C3A7, Canada

**Keywords:** model reduction, non-equilibrium thermodynamics, MaxEnt, dynamic MaxEnt, complex fluids, heat conduction, Ohm’s law

## Abstract

Any physical system can be regarded on different levels of description varying by how detailed the description is. We propose a method called Dynamic MaxEnt (DynMaxEnt) that provides a passage from the more detailed evolution equations to equations for the less detailed state variables. The method is based on explicit recognition of the state and conjugate variables, which can relax towards the respective quasi-equilibria in different ways. Detailed state variables are reduced using the usual principle of maximum entropy (MaxEnt), whereas relaxation of conjugate variables guarantees that the reduced equations are closed. Moreover, an infinite chain of consecutive DynMaxEnt approximations can be constructed. The method is demonstrated on a particle with friction, complex fluids (equipped with conformation and Reynolds stress tensors), hyperbolic heat conduction and magnetohydrodynamics.

## 1. Introduction

The problem of model reduction is ubiquitous in physics and mathematics. Consider a system (physical or mathematical) that can be regarded on two levels of description, upper (more detailed level) and lower (less detailed). The state variables on the lower level contain less information than state variables on the upper level. A projection from the upper level to the lower level is necessary to state the problem of model reduction correctly. Assume, moreover, that dynamics of the state variables on the upper level is granted, but one wishes to see evolution of the lower variables. The reason can be for instance the complexity of the detailed evolution, availability of experimental observations or simplicity of the lower description. A reduction of dynamics from the upper level to the lower (less detailed) level is called model reduction.

There is no general model reduction technique applicable to all systems. However, many physically based (we do not focus on formal mathematical expansion methods although we make a certain comparison below) methods have been developed, such as the Chapman–Enskog expansion [1] or other series expansions [2], projector operator techniques [3,4], or the method of natural projector, invariant slow manifolds, entropic scalar product and Ehrenfest reduction [5,6,7,8]. A common feature of the reduction techniques is the recognition of entropy, since entropy (measuring unavailable information) grows during the passage from the more detailed level to the less detailed. In particular, states on the higher level corresponding to maximum entropy are referred to as the quasi-equilibrium manifold. This manifold is constructed by maximization of entropy on the upper level while knowing the result of the projection of the state variables to the lower level. This is the principle of Maximum Entropy (MaxEnt), see e.g., [9,10].

Apart from the relations between state variables on the two levels of description, it is necessary to find relations between vector fields generating evolution on the two levels. Indeed, evolution equations on the higher level can be seen as a motion along a given vector field (following arrows of the field) while the corresponding vector field on the lower level is to be found. For instance, in the Ehrenfest reduction, the vector field on the higher level is first prolonged by Taylor series and subsequently projected to the lower level, using MaxEnt, and closed. See [7] for various geometric techniques on how to obtain the vector field on the lower level of description.

Our approach is different. First, we recognize not only the state variables, but also conjugate variables as independent quantities, being motivated by the formulation of thermodynamics in contact geometry [11,12,13]. Eventually, conjugate variables become related to the state variables through derivatives of a thermodynamic potential (e.g., energy or entropy), but they are considered independent from direct variables in contact formulation and exhibit evolution towards that relation while approaching a given level. State variables between the levels can be related by MaxEnt as in [7,9,14], but conjugate variables can be exploited to find the approximation of the vector field on the higher level of description such that evolution on the lower level becomes closed while extracting key features (“measured” by entropy potential) of the upper level dynamics. In other words, the vector field on the lower level becomes tangent to the quasi-equilibrium manifold, see [15] or [14]. We refer to this method as Dynamic MaxEnt (DynMaxEnt).

Novelty of this paper lies in the following points: (i) The DynMaxEnt is introduced in the energetic representation, which simplifies the calculations, (ii) a chain or higher order DynMaxEnt approximations is identified, (iii) DynMaxEnt is compared to asymptotic expansions and (iv) it is applied to a reduction of various complex continuum dynamics to classical irreversible thermodynamics (conformation tensor, Reynolds stress, hyperbolic heat conduction and magnetohydrodynamics).

In Section 2, we first recall the usual (static) Maximum Entropy principle (MaxEnt). In Section 3, we introduce the method of Dynamic MaxEnt, which serves as a reduction from the more detailed evolution equations to less detailed, and then we demonstrate it on damped particle dynamics. In Section 4, we use the DynMaxEnt method for reduction of dynamics of complex fluids, hyperbolic heat conduction and electromagnetohydrodynamics to the Navier–Stokes–Fourier system and magnetohydrodynamics. Finally, in Section 5, we present a geometric motivation for the DynMaxEnt method in the framework of contact geometry.

## 2. Static MaxEnt

As a thorough understanding of (static) maximum entropy (MaxEnt) method is required, we shall recapitulate its key steps, what it means and what it provides.

Let us denote the state variables on the more microscopic level as x∈M, M being the manifold (often vector space) of the state variables, with conjugate variables $x^{*}$ via entropy ${}^{{}^{↑}}S$. Furthermore, let us assume that there is a projection that relates two sets of variables x∈M, y∈N via π(x)=y where the latter corresponds to the more macroscopic level of description. The static MaxEnt provides an inverse mapping $\tilde{x}\left(y\right)$ such that $\tilde{x}\left(y\right)$ is the point of preimage $\pi^{-1}\left(y\right)$ with the highest entropy. This determines a manifold in M, the MaxEnt manifold $\tilde{M}$.

The inverse mapping can be achieved by maximization of entropy while keeping the constraint that π(x)=y, i.e., by the method of Lagrange multipliers. More geometrically, it can be also done via two consecutive Legendre transformations which together correspond to maximisation with a constraint y=π(x) [14,16]. In particular, as a first step, one obtains a relation $\tilde{x}\left(y^{*}\right)$ from solving
(1)$$0=\partial_{x}\underset{{}^{↑}\phi }{\underset{︸}{\left(-{}^{↑}S\left(x\right)+\langle y^{*},\pi \left(x\right)\rangle \right)}}$$
while the lower conjugate entropy is ${{}^{↓}S}^{*}\left(y^{*}\right)={}^{↑}\phi \left(\tilde{x}\left(y^{*}\right),y^{*}\right)$. Note that this (actually generalized) Legendre transformation can not be inverted.

Next, Legendre transformation of the lower conjugate entropy provides lower entropy and a relation $\tilde{y^{*}}\left(y\right)$:(2)$$0=\partial_{y^{*}}\underset{{}^{↓}\phi^{*}}{\underset{︸}{\left(-{{}^{↓}S}^{*}\left(y^{*}\right)+\langle y,y^{*}\rangle \right)}},$$
which solution yields $\tilde{y^{*}}\left(y\right)$ and ${}^{↓}S\left(y\right)={}^{↓}\phi^{*}\left(y,\tilde{y^{*}}\left(y\right)\right)$. Finally, the “inverse” mapping to the projection y=π(x) is $x=\tilde{x}\left(\tilde{y^{*}}\left(y\right)\right):=\pi^{-1}\left(y\right)$. Note that it is now evident that $\pi^{-1}$ is not a one-to-one mapping but rather a mapping identifying the most probable (with respect to the lower entropy) set of values of the microscale variable x that correspond to a given macroscale state y. We shall refer to this method as MaxEnt and see Figure 1 for a summary of the method.

## 3. Dynamic MaxEnt

A form of the Dynamic MaxEnt reduction first appeared in [15] in the context of contact geometry. Here, we further develop DynMaxEnt using the energetic representation, which simplifies the procedure so that reductions, for example, from complex fluid dynamics to Navier–Stokes equation and from hyperbolic heat conduction to the Fourier law are carried out easily. Moreover, we generalize the method to an infinite chain of consecutive approximations.

The key idea behind dynamic MaxEnt is to treat conjugate variables $x^{†}$ as independent quantities ensuring invariance of the quasi-equilibrium manifold. This step is motivated by contact geometry, where conjugate variables are indeed granted independence and evolve towards the corresponding Gibbs–Legendre manifold, see Section 6.1.

### 3.1. First Order DynMaxEnt Reduction

Here, we propose an extension to the static version of MaxEnt (This extension is based on our previous work in [14]; however, we clarify, extend and elaborate on this method in this article.), which provides in addition to the lower entropy ${}^{↓}S\left(y\right)$ and the inverse projection $\pi^{-1}\left(y\right)$ the dynamics on the lower level, i.e., reduced evolution equations.

Assuming the evolution on the more microscopic (higher) level can be expressed as
(3)$${\dot{x}}^{i}=V^{i}\left(x,x^{†}\right),$$
where $x^{†}$ are conjugate variables which can be eventually identified with derivatives of energy (being the choice in this paper or e.g., [17,18,19]), entropy or another thermodynamic potential. Let us denote the yet unspecified functional by ${}^{↑}\Phi$, i.e., $x_{i}^{†}={}^{↑}\Phi_{x^{i}}$. Components of the right-hand side of the evolution equations $V^{i}$ can be interpreted as elements of a vector field on the manifold of state variables M, x∈M. With the inverse relation (we shall use simply $\tilde{x}\left(y\right)$ instead of $\pi^{-1}\left(y\right)$ hereafter) at hand, which enslaves the microscale state variables x in terms of the macroscale (or reduced) variables y, we may evaluate the more detailed evolution equations on the MaxEnt manifold as
(4)$${\dot{\tilde{x}}}^{i}=\frac{\partial {\tilde{x}}^{i}}{\partial y^{a}}{\dot{y}}^{a}=V^{i}\left(\tilde{x}\left(y\right),x^{†}=y^{†}\cdot \frac{\partial \pi }{\partial x}{|}_{\tilde{x}\left(y\right)}\right),$$
noting that
(5)$$\frac{\partial {}^{{}^{↑}}\Phi (\tilde{x}\left(\pi \left(x\right)\right)}{\partial x^{i}}{|}_{\tilde{x}\left(y\right)}=\frac{\partial {}^{{}^{↓}}\Phi }{\partial y}\cdot \frac{\partial \pi }{\partial x}{|}_{\tilde{x}\left(y\right)}$$
for ${}^{↓}\Phi \left(y\right)={}^{↑}\Phi \left(\tilde{x}\left(y\right)\right)$. By projection π, we obtain the reduced equations
(6)$${\dot{y}}^{a}=\frac{\partial \pi^{a}}{\partial x^{i}}{|}_{\tilde{x}\left(y\right)}{\dot{\tilde{x}}}^{i}=\frac{\partial \pi^{a}}{\partial x^{i}}{|}_{\tilde{x}\left(y\right)}V^{i}\left(\tilde{x}\left(y\right),x^{†}=y^{†}\cdot \frac{\partial \pi }{\partial x}{|}_{\tilde{x}\left(y\right)}\right),$$
where it should be noted that the reduced equations have to live on the lower level. To this end, a relation ${\tilde{x}}^{†}\left(y,y^{†}\right)$ has to be identified. By comparing these last two evolution equations, it follows that
(7)$$\frac{\partial {\tilde{x}}^{i}}{\partial y^{a}}\frac{\partial \pi^{a}}{\partial x^{j}}{|}_{\tilde{x}\left(y\right)}V^{j}\left(\tilde{x}\left(y\right),x^{†}=y^{†}\cdot \frac{\partial \pi }{\partial x}{|}_{\tilde{x}\left(y\right)}\right)=V^{i}\left(\tilde{x}\left(y\right),x^{†}\left(y,y^{†}\right)\right),$$
which is a consistency condition entailing relations among conjugate variables ${\tilde{x}}^{†}\left(y,y^{†}\right)$. Without this condition, the evolution (Equation 4) would leave the manifold of MaxEnt states $\tilde{x}\left(y\right)$, the vector field would be sticking out of the MaxEnt manifold $\tilde{M}$. Condition (Equation 7) is actually an equation for $x^{†}$, the solution of which will be denoted by ${\tilde{x}}^{†}\left(y,y^{†}\right)$. After substitution into Equation (Equation 6), we obtain the reduced evolution equations for y, which is the result of the first order DynMaxEnt reduction.

The first order DynMaxEnt reduction can be summarized as the sequence
(8)$$y\overset{\mathrm{MaxEnt}}{\to }\tilde{x}\left(y\right)\overset{\mathrm{Equation}(7)}{\to }{\tilde{x}}^{†}\left(y,y^{†}\right),$$
which ends up in a closed system of equations for the reduced state variables y. The lower level potential follows from the projection as ${}^{↓}\Phi \left(y\right)={}^{↑}\Phi \left(\tilde{x}\left(y\right)\right)$.

Note, however, that the link tying x and $x^{†}$ was broken because ${\tilde{x}}^{†}$ was determined by solving Equation (Equation 7) instead of differentiating the potential ${}^{↑}\Phi$ with respect to x at $\tilde{x}\left(y\right)$. This leads to the higher order DynMaxEnt reduction.

### 3.2. Higher Order DynMaxEnt

From the first order DynMaxEnt, we have acquired independent relations among direct and conjugate variables, $\tilde{x}\left(y\right)$ and ${\tilde{x}}^{†}\left(y,y^{†}\right)$, but at the same time direct and conjugate variables are linked via the potential ${}^{↑}\Phi$. This entails that these two relations cannot be independent and some of the relations have to be violated.

There are two possible remedies of this situation both being iterative: (i) to correct the upper entropy so that the MaxEnt value is indeed a conjugate to the identified relation ${\tilde{x}}^{†}\left(y,y^{†}\right)$ on the lower level; however, this change in entropy means that MaxEnt has to be recalculated including everything that follows as well and the situation repeats itself; (ii) to correct the MaxEnt value of the direct variable so that the direct and conjugate values are now in line with the upper entropy evaluated on the lower level; however, this again modifies the dependent evolution equations on the lower level, in turn the value of upper conjugate variables and the situation repeats.

Although the former correction, the correction of the upper entropy, might seem natural, it turns out that it cannot be a general approach as the corrected entropy would fail to meet the requirement of being an even function with respect to time reversal when the reduced state variables change parity during the transition to a lower level, see Section A.1. We shall now develop the latter approach in detail (forming a possibly infinite chain of corrections) and compare it with asymptotic methods for upscaling evolution equations.

To repair the link between x and $x^{†}$, which remained broken after the first order DynMaxEnt reduction, an additional correction of the state variables x is required,
(9)$$\frac{\partial {}^{↑}\Phi }{\partial x}{|}_{{\tilde{x}}^{\left(2\right)}}={\tilde{x}}^{†}\left(y,y^{†}\right).$$

The solution to this equation gives the second iteration of the value of the state variable, ${\tilde{x}}^{\left(2\right)}\left(y,y^{†}\right)$.

However, is the consistency condition (Equation 7) satisfied at ${\tilde{x}}^{\left(2\right)}$? Not in general. Therefore, the condition can be regarded as an equation for the second-order iteration of $x^{†}$, ${\tilde{x}}^{†(2)}$. This iterative chain can be summarized as
(10)$$y\overset{\mathrm{MaxEnt}}{\to }\tilde{x}\left(y\right)\overset{\mathrm{Equation}(7)}{\to }{\tilde{x}}^{†}\left(y,y^{†}\right)\overset{\mathrm{Equation}(9)}{\to }{\tilde{x}}^{\left(2\right)}\left(y,y^{†}\right)\overset{\mathrm{Equation}(7)}{\to }{\tilde{x}}^{†(2)}\left(y,y^{†}\right)\to \cdots ,$$
which can continue indefinitely. We have thus found an infinite chain of DynMaxEnt reduction, which leads to evolution equations for the reduced state variables y in a closed form. Note that the lower potential corresponds to the chosen step *k* in the infinite chain of DynMaxEnt as ${}^{↓}\Phi^{\left(k\right)}\left(y\right)={}^{↑}\Phi \left({\tilde{x}}^{\left(k\right)}\left(y\right)\right)$.

It is now apparent that any correction in this infinite chain of DynMaxEnt reduction leads to another correction as any of the adaptations, be it in direct or conjugate variables or even entropy, yield new disparity. Where should one end the iteration and what is the best choice of a method generating evolution equations on the lower level of description while retaining the thermodynamic structure and knowledge of the equations?

First, it should be mentioned that the aim of dynamic MaxEnt is to arrive at reasonable evolution equations for a specified level of description. We conjecture that the static MaxEnt value of direct variables $\tilde{x}\left(y\right)$ is the best choice as it corresponds to the most probable, least biased relation [20] between direct lower and upper state variables. The conjugate variables then follow from the requirement of the dynamics being such that it does not drive the system away from the MaxEnt values of the direct variables, i.e., ${\tilde{x}}^{†}\left(y,y^{†}\right)$. Finally, the lower potential is chosen as the first correction ${}^{↓}\Phi^{\left(2\right)}\left(y\right)={}^{↑}\Phi \left({\tilde{x}}^{\left(2\right)}\left(y\right)\right)$. The reason is twofold: (i) a correction of the lower entropy neither affects the static MaxEnt value nor does it explicitly change the evolution equations that are given in terms of direct and conjugate variables (it affects them indirectly via a resulting change of the relation of conjugate to direct variables); (ii) the corrected entropy is more tightly linked to the evolution equations rather than to the static MaxEnt which we aim to extend. Additionally, as we shall see below, in the special case of projection corresponding to relaxation of fast variables, the ${}^{↓}S$ entropy is simply the upper entropy but where all the effects of fast variables are neglected while the correction ${{}^{↓}S}^{\left(2\right)}$ includes additional (typically non-local) effects resulting from this transition between levels.

In short, direct state variables are set-up in such a way that their most probable value (with given information about the system comprised in entropy) is always kept on the lower level (although it might evolve with the evolution of the lower state variables). The choice of conjugate variables entails persistence of the direct variables exactly on their most probable values. Lower entropy corresponds to the set-up of conjugate rather than direct variables, which contain some non-trivial effects due to the transition of scales.

### 3.3. Prototype Example—Damped Particle

Let us now demonstrate the DynMaxEnt reduction on a prototype example—damped particle in a potential field. The state variables represent position, momentum and entropy of the particle, x=(r,p,s), and momentum is considered as the fast variable, which is to be reduced, i.e., y=(r,s). The projection to the lower (less detailed) level is thus π:(r,p,s)→(r,s).

The more detailed evolution equations are ([14], Ch 5.3.1)
(11a)$$\begin{array}{ccc} \dot{r} & = & p^{†}, \end{array}$$
(11b)$$\begin{array}{ccc} \dot{p} & = & -r^{†}-\frac{1}{\tau }\frac{p^{†}}{s^{†}}, \end{array}$$
(11c)$$\begin{array}{ccc} \dot{s} & = & \frac{1}{\tau }{\left(\frac{p^{†}}{s^{†}}\right)}^{2}, \end{array}$$
where $x^{†}$ can be interpreted as conjugates with respect to energy
(12)$${}^{↑}e=\frac{p^{2}}{2m}+V\left(r\right)+\varepsilon \left(s\right),$$
consisting of kinetic energy, potential energy and internal energy. The evolution equations represent Hamilton canonical equations (for r and p) equipped with friction in p and entropy production, c.f. [21]. Total energy is conserved by the evolution equations, $\dot{e}=r^{†}\dot{r}+p^{†}\dot{p}+s^{†}\dot{s}=0$.

#### 3.3.1. First Order DynMaxEnt

From relation (Equation 12), it follows that entropy on the higher level reads
(13)$${}^{↑}s=s\left(e-\frac{p^{2}}{2m}-V\left(r\right)\right),$$
s(•) being the inverse function to ε(•). Entropy attains maximum (for a given energy and position) at $\tilde{p}=0$, which is the MaxEnt value of momentum.

The consistency condition (Equation 7) becomes in this particular case
(14)$$\left(\begin{array}{cc} 1 & 0\\ 0 & 0\\ 0 & 1 \end{array}\right)\cdot \left(\begin{array}{ccc} 1 & 0 & 0\\ 0 & 0 & 1 \end{array}\right)\cdot \left(\begin{array}{c} p^{†}\\ -r^{†}-\frac{1}{\tau }\frac{p^{†}}{r^{†}}\\ \frac{1}{\tau }{\left(\frac{p^{†}}{r^{†}}\right)}^{2} \end{array}\right)=\left(\begin{array}{c} p^{†}\\ -r^{†}-\frac{1}{\tau }\frac{p^{†}}{s^{†}}\\ \frac{1}{\tau }{\left(\frac{p^{†}}{r^{†}}\right)}^{2} \end{array}\right),$$
which can be simplified to
(15)$$0=-r^{†}-\frac{1}{\tau }\frac{p^{†}}{s^{†}}.$$

The solution to this equation is the first iteration of the conjugate momentum, ${\tilde{p}}^{†}=-\tau s^{†}r^{†}$. Plugging this back into the evolution equations leads to
(16a)$$\begin{array}{ccc} \dot{r} & = & -\tau s^{†}r^{†}, \end{array}$$
(16b)$$\begin{array}{ccc} \dot{s} & = & \tau {\left(r^{†}\right)}^{2}. \end{array}$$

These are the reduced equations obtained by the first order DynMaxEnt procedure. The † symbols denote derivative of a yet unspecified energy on the lower level of description ${}^{↓}e\left(r,s\right)$. Note that the energy is again conserved and that entropy is produced. To make the reduced evolution equations closed, we should identify the lower energy ${}^{↓}e$ as the MaxEnt value of the higher-level energy,
(17)$${}^{↓}e={}^{↑}e\left(r,\tilde{p},s\right)=V\left(r\right)+\varepsilon \left(s\right).$$

Equations (16) then gain an explicit form manifesting that the particle tends to the minimum of the potential *V* while producing entropy.

#### 3.3.2. Second Order DynMaxEnt

By solving equation (Equation 14), we have actually broken the link between p and $p^{†}$, namely $p^{†}\neq \partial {}^{↑}e/\partial p$. To recover that link, we have to find a new value of p that corresponds to the obtained ${\tilde{p}}^{†}$,
(18)$$\frac{{\tilde{p}}^{\left(2\right)}}{m}=\frac{\partial {}^{↑}e}{\partial p}{|}_{{\tilde{p}}^{\left(2\right)}}=-\tau \frac{\partial {}^{↑}e}{\partial s}\frac{\partial {}^{↑}e}{\partial r}=-\tau \frac{\partial \varepsilon }{\partial s}\frac{\partial V}{\partial r}.$$

Plugging this new value of p into energy ${}^{↑}e$ gives a new energy on the lower level,
(19)$${{}^{↓}e}^{\left(2\right)}=\frac{m}{2}{\left(\tau \frac{\partial \varepsilon }{\partial s}\frac{\partial V}{\partial r}\right)}^{2}+V\left(r\right)+\varepsilon \left(s\right),$$
which can be seen as a weakly nonlocal correction of the MaxEnt lower energy ${}^{↓}e$.

Once the link between p and $p^{†}$ through derivative of energy (Legendre transformation) has been recovered by updating $\tilde{p}$ to ${\tilde{p}}^{\left(2\right)}$, condition (Equation 14) no longer holds. To satisfy the condition, we have to find a new value of the conjugate momentum by solving (assuming that τ∂ε/∂s=ζ=const for simplicity)
(20)$$\begin{array}{ccc} {\dot{\tilde{p}}}^{\left(2\right)} & = & -m\zeta \frac{\partial^{2}V}{\partial r^{2}}\dot{r}=-m\zeta \frac{\partial^{2}V}{\partial r^{2}}{\tilde{p}}^{†(2)}\\ & = & -r^{†}-\frac{1}{\zeta }{\tilde{p}}^{†(2)}=-\frac{\partial V}{\partial r}-\frac{1}{\zeta }{\tilde{p}}^{†(2)}. \end{array}$$

The solution to this equation is
(21)$${\tilde{p}}^{†(2)}=\frac{-\zeta \frac{\partial V}{\partial r}}{1-m\zeta^{2}\frac{\partial^{2}V}{\partial r^{2}}}∼-\zeta \frac{\partial V}{\partial r}-m\zeta^{3}\frac{\partial V}{\partial r^{2}}+O{\left(\zeta \right)}^{5}.$$

The latter contribution is obviously of the second order in the relaxation time τ while the former contribution is identical to the ${\tilde{p}}^{†}$ found above. This second-order correction of conjugate momentum can be plugged back into the equations for $\dot{r}$ and $\dot{s}$ to obtain
(22a)$$\begin{array}{ccc} \dot{r} & = & \frac{-\zeta \frac{\partial V}{\partial r}}{1-m\zeta^{2}\frac{\partial^{2}V}{\partial r^{2}}}, \end{array}$$
(22b)$$\begin{array}{ccc} \dot{s} & = & \frac{\zeta }{s^{†}}\frac{{\left(\frac{\partial V}{\partial r}\right)}^{2}}{{\left(1-m\zeta^{2}\frac{\partial^{2}V}{\partial r^{2}}\right)}^{2}}, \end{array}$$
which is the second-order DynMaxEnt reduction. It is instructive to compare asymptotic expansion of the evolution equations for small τ/ζ with DynMaxEnt.

Note that the singularity suggested by this second order DynMaxEnt is not physically relevant but rather an invitation to a higher order as Equation (Equation 20) reveals—independence of $p^{†(2)}$ and $\partial_{r}V=0$. In addition, if τ≪1, i.e., ζ≪1 is small, DynMaxEnt generates a converging sequence of evolution laws where there is no blow-up. Hence, when the parameter ζ or τ is not small, the “corrections” stemming from higher order DynMaxEnt might be large and significant. Hence, as we shall further explore below, the higher order corrections seem to be mainly relevant when a small parameter is present.

#### 3.3.3. Damped Particle by Asymptotic Expansions

A hallmark example of the usage of asymptotic expansions in statistical physics is the Chapman–Enskog expansion in kinetic theory [2,22].

In general, the two approaches, asymptotic expansion vs. Dynamic MaxEnt, are very different concepts of obtaining description of a system on a coarser level. The asymptotic analysis (such as the method of multiple scales, homogenisation, regular and singular perturbation methods or boundary layer analysis) does not explicitly assess the lower level of description or what exactly does the asymptotic approximation represents, but, from intuition and the dimensionless form of the evolution equations, one can use the presence of a small parameter to relate two levels of description (more precisely, two different spatial or temporal scales). The number of state variables and evolution equations are typically not changed (although in Chapman–Enskog analysis, evolution equations for several moments of the distribution function somewhat naturally appear) and a suitable choice of the form of expansion series allows a nested problem formulation on the coarser level by sequentially solving each asymptotic order. Note, however, that there is not a unique way to include the small parameter in spatial scaling, etc.

Dynamic MaxEnt, on the other hand, uses projections to the lower level of description (e.g., by relaxation of fast variables). In particular, the number of evolution equations (state variables) can be very different on the two levels in contrast to asymptotic analysis.

Let us compare the asymptotic method of solution with the Dynamic MaxEnt method on the simple problem of a damped particle where the dimensionless parameter 1/τ measures the ratio of the reversible and irreversible evolution. Typically, the timescale of relaxation of dissipative processes is much shorter than the remaining evolution, hence τ≪1. The solution to the problem can be searched in the form of asymptotic expansions
(23)$$p=p_{0}+\tau p_{1}+\tau^{2}p_{2}+O\left(\tau^{3}\right),\;r=r_{0}+\tau r_{1}+O\left(\tau^{2}\right),\;s=s_{0}+\tau s_{1}+O\left(\tau^{2}\right)$$
and the conjugate $r^{†}$ variable has the following expansion
$$r^{†}=V_{r}\left(r_{0}+\tau r_{1}+\tau^{2}r_{2}\right)=V_{r}\left(r_{0}\right)+\tau r_{1}V_{r}\left(r_{o}\right)+O\left(\tau^{2}\right).$$

Noting that $p^{†}=p/m$, the leading order solution is
$$\tau^{-1}:\;p_{0}=0,$$
while the first subleading order gives
$$\tau^{0}:p_{1}=-ms^{†}V_{r}\left(r_{0}\right),\;{\dot{r}}_{0}=\frac{p_{0}}{m}=0.$$

Therefore, $p_{1}$ is independent of time and the second subleading order reads
(24)$$p_{2}=-ms^{†}V_{rr}\left(r_{0}\right)r_{1}-ms^{†}{\dot{p}}_{1}=-ms^{†}V_{rr}\left(r_{0}\right)r_{1},\;{\dot{r}}_{1}=\frac{p_{1}}{m}=-s^{†}V_{r}\left(r_{0}\right),$$
while, finally, ${\dot{r}}_{2}=p_{2}/m$. In summary, the reduced evolution up to the second order can be characterised by the following set of equations:(25)$${\dot{r}}_{0}=0,\;{\dot{r}}_{1}=-s^{†}V_{r}\left(r_{0}\right),\;{\dot{r}}_{2}=-s^{†}V_{rr}\left(r_{0}\right)r_{1}.$$

From the first subleading order, one could estimate that perhaps $p_{0}=0,\;p_{1}=-ms^{†}r^{†}$ and thus $\dot{r}=-\tau ms^{†}r^{†}$. We can now straightforwardly verify this by using the asymptotic expansions (Equation 23).

Hence, indeed, we have $p=-\tau ms^{†}r^{†}$ while the remaining equations can be written in the following way:
(26a)$$\dot{r}=-\tau s^{†}r^{†},$$
(26b)$$\dot{s}=\tau {\left(r^{†}\right)}^{2},$$
which means that the particle tends to the minimum of the potential V(r). These equations are in line with Equations (16), obtained by DynMaxEnt. Additionally, the leading order solution in the asymptotic method corresponds to MaxEnt $\tilde{x}\left(y\right)$ value (as in the Chapman–Enskog solution to the Boltzmann equation where the leading order solution corresponding to vanishing collision term is the local Maxwell distribution).

Note that we proposed the DynMaxEnt method to be the first order in state variables (both direct and conjugate) accompanied by the second order lower energy ${}^{↓}e^{\left(2\right)}$. In particular, if we now compare the lower evolution equations from the asymptotic (26) and dynamic MaxEnt (16), we observe that they are identical with the exception stemming from the correction of the lower energy (conjugates in the lower evolution equations are always with respect to the lower energy). Furthermore, the comparison of the asymptotic solution p, Equation (24), and of the dynamic MaxEnt result reveals that: (i) the structure of dynamic MaxEnt iteration resembles asymptotic expansion for a small parameter τ; (ii) the leading order solutions are the same, however, the subleading terms differ.

#### 3.3.4. Relation to GENERIC

Evolution equations for the damped particle (11) can be seen as a particular realization of the General Equation for Non-Equilibrium Reversible-Irreversible Coupling (GENERIC) [4,14,23,24], with the following building blocks
(27)$$L=\left(\begin{array}{ccc} 0 & 1 & 0\\ -1 & 0 & 0\\ 0 & 0 & 0 \end{array}\right)\;\mathrm{and}\;\Xi^{\left(p\right)}=\frac{1}{2}\frac{1}{\tau }{\left(p^{*}\right)}^{2},$$
where the former expression defines a Poisson bivector while the latter dissipation potential (Irreversible evolution generated by a dissipation potential is also called gradient dynamics. It is in tight relation to the Steepest Entropy Ascent [25,26], which is essentially equivalent to the formulation of GENERIC with dissipative brackets [4].). The dissipation potential is naturally formulated in terms of $p^{*}$, which can be interpreted as derivative of entropy. The evolution equation consists of reversible part L·dE and irreversible $\Xi_{p^{*}}{|}_{p^{*}=S_{p}}$. In order to use only derivative w.r.t. energy, we recall the relation among the two representations $p^{*}=-p^{†}/s^{†}$, see [14,27] for more details (Indeed, on the Gibbs–Legendre manifold, where conjugate variables are identified with derivatives of thermodynamic potentials, one has ${\left(\frac{\partial s}{\partial p_{i}}\right)}_{e}=-{\left(\frac{\partial e}{\partial p_{i}}\right)}_{s}{\left(\frac{\partial e}{\partial s}\right)}_{p}^{-1}$ and ${\left(\frac{\partial e}{\partial s}\right)}_{p}={\left(\frac{\partial s}{\partial e}\right)}_{p}^{-1}$.) The equations are then explicitly
(28)$$\frac{d}{dt}\left(\begin{array}{c} r\\ p\\ s \end{array}\right)=\left(\begin{array}{ccc} 0 & 1 & 0\\ -1 & 0 & 0\\ 0 & 0 & 0 \end{array}\right)\cdot \left(\begin{array}{c} r^{†}\\ p^{†}\\ s^{†} \end{array}\right)+\left(\begin{array}{c} 0\\ -\frac{1}{\tau }\frac{p^{†}}{s^{†}}\\ \frac{1}{\tau }{\left(\frac{p^{†}}{s^{†}}\right)}^{2} \end{array}\right),$$
which is the same as Equations (11). The equations thus possess the GENERIC structure.

After the relaxation of the momentum p, expressed by Equation (Equation 15), evolution of positions becomes irreversible, given by Equation (16). Is this equation compatible with GENERIC? In other words, is there a dissipation potential generating that evolution? The answer is affirmative and actually easy to find at least in the case of the damped particle. Indeed, by evaluating dissipation potential $\Xi^{\left(p\right)}$ at the constitutive relation (Equation 15), we obtain dissipation potential for the lower level
(29)$$\Xi^{\left(r\right)}\left(r^{*}\right)=\Xi^{\left(p\right)}{|}_{p^{*}=-\tau r^{†}=\tau \frac{r^{*}}{e^{*}}}=\frac{\tau }{2}{\left(\frac{r^{*}}{e^{*}}\right)}^{2},$$
where $e^{*}=S_{e}=1/s^{†}$, *e* being total energy of the particle. Derivative of this dissipation potential w.r.t. $p^{*}$ reads
(30)$$\dot{r}=\Xi_{r^{*}}^{\left(r\right)}=\tau \frac{r^{*}}{{\left(e^{*}\right)}^{2}}=-\tau s^{†}r^{†},$$
which is the reduced evolution Equation (16). In other words, at least in this particular case the reduced evolution equation for r is generated by a dissipation potential constructed from the original dissipation potential for p by a projection, see also [21]. At least in the case of damped particle, the reduced evolution is a particular realization of GENERIC, and the reduced dissipation potential is obtained from the original dissipation potential.

### 3.4. Summary of the Dynamic MaxEnt Method

A few remarks should be pointed out:MaxEnt with constraints corresponding to a chosen projection linking the two levels of description is equivalent to MinEne with the same constraints. This follows from the same argument as used in ([27], Ch 5.1) only with the addition of constraints which, crucially, describe the same manifold in the phase space of (S,E,x).It is important to consider direct and conjugate variables to be independent until reaching the Legendre manifold where they are related via appropriate entropy. This stands out during the MaxEnt procedure as noted above and in the contact-geometric formulation in Section 5.Static MaxEnt should be seen as not providing a relation among conjugate variables but rather only among the direct ones. This can be appreciated in a special case of projection that corresponds to a relaxation of fast variables in the system, hence removing some of the state variables x when a transition to a less microscopic description is carried out. For this point of view, it is essential to consider direct and conjugate variables as independent.If direct and conjugate variables are not considered independent during MaxEnt and hence MaxEnt would provide MaxEnt values for both reduced and conjugate variables, the dynamics on MaxEnt manifold would be (typically) such that its vector field would be “sticking out”, i.e., driving the dynamics out of the MaxEnt manifold. The correction in finding ${\tilde{x}}^{*}\left(y,y^{*}\right)$ is exactly such that the MaxEnt manifold $\tilde{M}$ in direct variables is never left by the dynamics.The distinction of direct and conjugate variables has several crucial benefits. First, it guarantees the thermodynamic structure of the evolution equations (potentials, reversible and irreversible parts of equations, CR-GENERIC or possibly GENERIC structure). Secondly, this distinction of state variables enables to always sustain the MaxEnt, i.e., the most probable, value of the direct variables on the lower description (as discussed above). This cannot be achieved in an asymptotic framework/approach, where such knowledge is not at hand and the solution is searched in the form of asymptotic corrections to a leading order solution.Parity of the state variables with respect to time reversal typically changes during the DynMaxEnt reduction. For instance, the momentum, which was initially an odd variable, becomes proportional to the gradient of potential, which is even, or the conformation tensor (which is initially even) becomes proportional to the shear rate, which is odd. Similarly, behavior with respect to space-time transformations (e.g., Galilean boost) also changes. For instance, momentum of a particle, the value of which of course depends on the choice of inertial laboratory frame, becomes independent of Galilean boosts after the reduction (being proportional to gradient of the potential). This is compatible with the multiscale point of view of the studied systems. For instance, if one is able to measure the conjugate entropy flux w (see Section 4.3) directly, one can also see that Galilean boost affects it, and one should stay on the level of description where w is among the state variables. On the other hand, if one only measures w in the relaxed state, where it is proportional to the gradient of temperature, one does not see the effect of Galilean boosts anymore, and one can safely work on the Fourier level, where w is no longer among the state variables. In summary, the behavior of physical quantities with respect to time reversal and space-time transformations crucially depends on the chosen level of description, and this level-dependent behavior is compatible with the multiscale point of view of physical systems.

The comparison to the asymptotic expansion methods provided further insight into the Dynamic MaxEnt method. For example, the asymptotic analysis provides corrections to the solution at each order, whereas Dynamic MaxEnt provides relations for closures/fluxes. In the particular examples above, it may seem that both methods yield the same set of leading order evolution equations but note that this is only when one can explicitly solve the asymptotic problems in each order. As noted above, typically asymptotic methods do not change the number of equations as opposed to Dynamic MaxEnt, where the number is given by the projection. Asymptotic methods also typically do not change parity with respect to time reversal, whereas DynMaxEnt does [28]. Finally, note that Dynamic MaxEnt does not rely per se on the presence of a small parameter but rather on the existence of a projection which is similar but not the same.

In summary, Dynamic MaxEnt does not have (and perhaps should not have) an aim to find other than the leading order evolution equations on a specified level. From all the observations and discussion above, we propose that the most appropriate choice for this aim is to find and keep the static MaxEnt values of direct variables exactly while identifying the conjugate state variables in such a way that evolution (dynamics) stays exactly on this MaxEnt value of direct variables; entropy is modified in such a way as to correspond to conjugate variables and the first correction of the direct variables. It seems that, if the zero-th order asymptotic solution coincides with the static MaxEnt value of the direct state variable $\tilde{x}\left(y\right)$, the asymptotic expansion method yields the same leading order behaviour as the Dynamic MaxEnt (although without the thermodynamic structure and new phenomena linked to the change of entropy accompanying the transition of scales). Interestingly, we found in the studied example that even higher order asymptotic solutions seem to be related to the Dynamic MaxEnt method, although it is yet unclear how exactly and will be subjected to further research.

## 4. Applications in Continuum Thermodynamics

Let us now demonstrate the Dynamic MaxEnt on a few examples in continuum thermodynamics, namely on relaxation of conformation tensor, relaxation of Reynolds stress, and hyperbolic heat conduction. The former two examples were already discussed in [14].

### 4.1. Suspension of Hookean Dumbbells

#### 4.1.1. Non-Equilibrium Thermodynamics of Conformation Tensor

First, let the state variables be density of matter ρ, momentum density u, entropy density *s* and conformation tensor $c^{ij}$, which expresses correlations of dumbbell orientation and prolongation. Poisson bracket expressing kinematics of these state variables is
(31)$$\begin{array}{ccc} {\left\{A,B\right\}}^{\left(c\right)} & = & {\left\{A,B\right\}}^{\left(FM\right)}+\int drc_{ij}\left(\partial_{k}A_{c_{ij}}B_{u_{k}}-\partial_{k}B_{c_{ij}}A_{u_{k}}\right)\\ & & +\int drc_{ij}\left(\left(A_{c_{kj}}+A_{c_{jk}}\right)\partial_{i}B_{u_{k}}-\left(B_{c_{kj}}+B_{c_{jk}}\right)\partial_{i}A_{u_{k}}\right), \end{array}$$
where $c^{ij}$ was identified with $c_{ij}$ for simplicity of notation. Bracket ${\left\{•,•\right\}}^{\left(FM\right)}$ is the fluid mechanics Poisson bracket (Equation A3). Note that the conformation tensor is related to the left Cauchy–Green tensor B by c=ρB. Reversible evolution equations for state variables x=(ρ,u,s,c) implied by bracket (Equation 29) are
(32a)$$\begin{array}{ccc} \frac{\partial \rho }{\partial t} & = & -\partial_{i}\left(\rho E_{u_{i}}\right), \end{array}$$
(32b)$$\begin{array}{ccc} \frac{\partial u_{i}}{\partial t} & = & -\partial_{j}\left(u_{i}E_{u_{j}}\right)-\rho \partial_{i}E_{\rho }-u_{j}\partial_{i}E_{u_{j}}-s\partial_{i}E_{s}-\\ & & -c_{jk}\partial_{i}E_{c^{jk}}+\partial_{k}\left(c_{kj}\left(E_{c_{ij}}+E_{c_{ji}}\right)\right), \end{array}$$
(32c)$$\begin{array}{ccc} \frac{\partial s}{\partial t} & = & -\partial_{i}\left(sE_{u_{i}}\right), \end{array}$$
(32d)$$\begin{array}{ccc} \frac{\partial c_{ij}}{\partial t} & = & -\partial_{k}\left(c_{ij}E_{u_{k}}\right)+c_{kj}\partial_{k}E_{u_{i}}+c_{ki}\partial_{k}E_{u_{j}}, \end{array}$$
where *E* is the total energy of the system.

With energy
(33)$$E=\int dr\left[\frac{u^{2}}{2\rho }+\varepsilon \left(\rho ,s,c\right)\right],$$
where internal energy ε still remains unspecified, evolution Equation ([Disp-formula FD32d-entropy-21-00715]) can be rewritten in terms of the upper-convected time-derivative as
(34)$$\overset{\nabla }{c}=-c\nabla \cdot v,\;v=\frac{u}{\rho },$$
which is the reversible part of Maxwell rheological model [29].

Considering a suspension of Hookean dumbbells, entropy is
(35)$$S=\int dr\left[\frac{1}{2}nk_{B}\ln\det c+s\left(n,e-\frac{u^{2}}{2\rho }-\frac{1}{2}H\;\mathrm{Tr}c\right)\right],$$
as derived for instance in [14]. Parameter *H* is the spring constant of the dumbbells, and *n* represents concentration of dumbbells. Derivative of this entropy with respect to c is
(36)$$c_{ij}^{*}=\frac{\partial S}{\partial c_{ij}}=\frac{1}{2}k_{B}nc_{ij}^{-1}-\frac{1}{2}\frac{H}{T}\delta_{ij},$$
which is equal to zero for
(37)$$\tilde{c}=\frac{k_{B}Tn}{H}I.$$

This is the MaxEnt value of c. Note that inverse temperature $T^{-1}$ is identified as derivative of entropy with respect to energy density.

Dissipation potential can be prescribed as
(38)$$\Xi^{\left(c\right)}=\int dr\Lambda_{c}c_{ij}c_{ik}^{*}c_{jk}^{*},$$
the derivative of which is
(39)$$\Xi_{c_{ij}^{*}}^{\left(c\right)}=2\Lambda_{c}c_{ik}c_{kj}^{*}.$$

The evolution equation of c then gains an irreversible term,
(40)$$\frac{\partial \Xi^{\left(c\right)}}{\partial c_{ij}^{*}}{|}_{c_{*}=S_{c}}=-\frac{\Lambda_{c}H}{T}\left(c_{ij}-\frac{k_{B}Tn}{H}\delta_{ij}\right).$$

After the transformation to the energetic representation (conjugates with respect to energy rather than entropy), the sum of the reversible and irreversible contributions to evolution equations for (ρ,u,c,s) is prepared for the DynMaxEnt reduction.

#### 4.1.2. DynMaxEnt to Hydrodynamics

Let us now apply the Dynamic MaxEnt reduction so that the conformation tensor c relaxes. The energetic representation reversible Equations (32) and irreversible Equation (Equation 40) together are
(41a)$$\begin{array}{ccc} \frac{\partial \rho }{\partial t} & = & -\partial_{i}\left(\rho u_{i}^{†}\right), \end{array}$$
(41b)$$\begin{array}{ccc} \frac{\partial u_{i}}{\partial t} & = & -\partial_{j}\left(u_{i}u_{j}^{†}\right)-\rho \partial_{i}\rho^{†}-u_{j}\partial_{i}u_{j}^{†}-s\partial_{i}s^{†}-\\ & & -c_{jk}\partial_{i}c_{jk}^{†}+\partial_{k}\left(c_{kj}\left(c_{ij}^{†}+c_{ji}^{†}\right)\right), \end{array}$$
(41c)$$\begin{array}{ccc} \frac{\partial s}{\partial t} & = & -\partial_{i}\left(su_{i}^{†}\right)+2\frac{\Lambda_{c}}{{\left(s^{†}\right)}^{2}}c_{ij}^{†}c_{ik}c_{kj}^{†}, \end{array}$$
(41d)$$\begin{array}{ccc} \frac{\partial c_{ij}}{\partial t} & = & -\partial_{k}\left(c_{ij}u_{k}^{†}\right)+c_{kj}\partial_{k}u_{i}^{†}+c_{ki}\partial_{k}u_{j}^{†}-\frac{2\Lambda_{c}}{s^{†}}c_{ik}c_{kj}^{†}. \end{array}$$

Consider now the isothermal incompressible case (The compressible case and the origin of incompressibility were discussed in [19].), i.e., $S_{e}=T=\mathrm{const}$, n=const and $\nabla \cdot u^{†}=0$. Equations ([Disp-formula FD41b-entropy-21-00715]) and ([Disp-formula FD41d-entropy-21-00715]) at the MaxEnt value of c (given by Equation (Equation 37)) become
(42a)$$\begin{array}{ccc} \frac{\partial u_{i}}{\partial t} & = & -\partial_{j}\left(u_{i}u_{j}^{†}\right)-\rho \partial_{i}\rho^{†}-u_{j}\partial_{i}u_{j}^{†}-s\partial_{i}s^{†}-\\ & & -\frac{k_{B}Tn}{H}\partial_{i}\mathrm{Tr}c^{†}+\frac{k_{B}Tn}{H}\partial_{k}\left(c_{ik}^{†}+c_{ki}^{†}\right), \end{array}$$
(42b)$$\begin{array}{ccc} 0 & = & \partial_{j}u_{i}^{†}+\partial_{i}u_{j}^{†}-2\frac{\Lambda_{c}}{T}c_{ij}^{†}. \end{array}$$

The last equation has a solution
(43)$$c^{†}=\frac{T}{2\Lambda_{c}}\left(\nabla u^{†}+{\left(\nabla u^{†}\right)}^{T}\right),\;\mathrm{Tr}c^{†}=0.$$

By plugging this solution into the equation for momentum density, we obtain the Navier–Stokes equation for momentum density
(44)$$\begin{array}{ccc} \frac{\partial u_{i}}{\partial t} & = & -\partial_{j}\left(u_{i}u_{j}^{†}\right)-\rho \partial_{i}\rho^{†}-u_{j}\partial_{i}u_{j}^{†}-s\partial_{i}s^{†}\\ & & +\frac{k_{B}T^{2}n}{H\Lambda_{c}}\partial_{k}\left(\partial_{i}u_{k}^{†}+\partial_{k}u_{i}^{†}\right), \end{array}$$
where the coefficient $k_{B}T^{2}n/H\Lambda_{c}$ corresponds to the shear viscosity and $u^{†}=E_{u}=v=u/\rho$ is the velocity.

The Dynamic MaxEnt reduction of the conformation tensor leads to the Newtonian shear stress tensor.

### 4.2. Reynolds Stress

We now turn to a complex fluid where momentum–momentum correlations between two neighboring particles matter, the correlation being expressed by the Reynolds stress tensor $R_{ij}$.

#### 4.2.1. Non-Equilibrium Thermodynamics of Reynolds Stress

Let the state variables be x=(ρ,u,s,R). The Poisson bracket expressing kinematics of x is
(45)$$\begin{array}{ccc} {\left\{A,B\right\}}^{\left(R\right)} & = & {\left\{A,B\right\}}^{\left(FM\right)}+\int drR_{ij}\left(\partial_{k}A_{R_{ij}}B_{u_{k}}-\partial_{k}B_{R_{ij}}A_{u_{k}}\right)-\\ & & -\int drR_{ij}\left(\left(A_{R_{kj}}+A_{R_{jk}}\right)\partial_{k}B_{u_{i}}-\left(B_{R_{kj}}+B_{R_{jk}}\right)\partial_{k}A_{u_{i}}\right), \end{array}$$
see, e.g., [14,30]. The reversible evolution equations are then
(46a)$$\begin{array}{ccc} \frac{\partial \rho }{\partial t} & = & -\partial_{i}\left(\rho E_{u_{i}}\right), \end{array}$$
(46b)$$\begin{array}{ccc} \frac{\partial u_{i}}{\partial t} & = & -\partial_{j}\left(u_{i}E_{u_{j}}\right)-\rho \partial_{i}E_{\rho }-u_{j}\partial_{i}E_{u_{j}}-s\partial_{i}E_{s}-\\ & & -R_{kj}\partial_{i}E_{R_{kj}}-\partial_{k}\left(R_{ij}\left(E_{R_{kj}}+E_{R_{jk}}\right)\right), \end{array}$$
(46c)$$\begin{array}{ccc} \frac{\partial s}{\partial t} & = & -\partial_{i}\left(sE_{u_{i}}\right), \end{array}$$
(46d)$$\begin{array}{ccc} \frac{\partial R_{ij}}{\partial t} & = & -\partial_{k}\left(R_{ij}E_{u_{k}}\right)-R_{kj}\partial_{i}E_{u_{k}}-R_{ki}\partial_{j}E_{u_{k}}. \end{array}$$

Let entropy be given by
(47)$$S^{\left(R\right)}=\int dr\left[s\left(\rho ,e-\frac{u^{2}}{2\rho }-\frac{1}{2m}\mathrm{Tr}R\right)+\frac{1}{2}k_{B}\frac{\rho }{m}\ln\det\frac{R}{Q_{R}}\right],$$
where $Q_{R}$ is an appropriately chosen constant. The reason for this entropy is analogical to the entropy (Equation 35). A derivative of entropy (Equation 47) with respect to R is
(48)$$R_{ij}^{*}=\frac{\partial S^{\left(R\right)}}{\partial R_{ij}}=-\frac{1}{2m}S_{e}\delta_{ij}+\frac{k_{B}\rho }{2m}{\left(R^{-1}\right)}_{ij},$$
which is equal to zero if and only if
(49)$$R_{ij}=k_{B}T\rho \delta_{ij}.$$

This is the MaxEnt value of R, at which the Reynolds stress is proportional to the unit matrix.

Similarly as in the preceding section, we choose dissipation potential
(50)$$\Xi^{\left(R\right)}=\int dr\Lambda_{R}R_{ij}^{*}R_{jk}R_{ki}^{*},$$
the derivative of which is
(51)$$\Xi_{R_{ij}^{*}}^{\left(R\right)}=2\Lambda_{R}R_{jk}R_{ki}^{*}.$$

Note that $R^{ij}$ was identified with $R_{ij}$ and similarly for $R^{*}$ for simplicity of notation. Evolution equation of R then gains an irreversible term
(52)$$\frac{\partial \Xi^{\left(R\right)}}{\partial R_{ij}^{*}}{|}_{R^{*}=S_{R}}=-\frac{\Lambda_{R}}{Tm}\left(R-k_{B}T\rho I\right).$$

By combining the reversible evolution (46) and irreversible (Equation 52), we obtain
(53a)$$\begin{array}{ccc} \frac{\partial \rho }{\partial t} & = & -\partial_{i}\left(\rho E_{u_{i}}\right), \end{array}$$
(53b)$$\begin{array}{ccc} \frac{\partial u_{i}}{\partial t} & = & -\partial_{j}\left(u_{i}E_{u_{j}}\right)-\rho \partial_{i}E_{\rho }-u_{j}\partial_{i}E_{u_{j}}-s\partial_{i}E_{s}-\\ & & -R_{kj}\partial_{i}E_{R_{kj}}-\partial_{k}\left(R_{ij}\left(E_{R_{kj}}+E_{R_{jk}}\right)\right), \end{array}$$
(53c)$$\begin{array}{ccc} \frac{\partial s}{\partial t} & = & -\partial_{i}\left(sE_{u_{i}}\right)+2\frac{\Lambda_{R}}{{\left(E_{s}\right)}^{2}}E_{R_{ij}}R_{jk}E_{R_{ki}}, \end{array}$$
(53d)$$\begin{array}{ccc} \frac{\partial R_{ij}}{\partial t} & = & -\partial_{k}\left(R_{ij}E_{u_{k}}\right)-R_{kj}\partial_{i}E_{u_{k}}-R_{ki}\partial_{j}E_{u_{k}}\\ & & -2\frac{\Lambda_{R}}{E_{s}}R_{jk}E_{R_{ki}}, \end{array}$$
which are the GENERIC evolution equations for fluid mechanics with Reynolds stress. For instance, the last equation can be rewritten as
(54)$$\begin{array}{ccc} \frac{\partial R_{ij}}{\partial t} & = & -\partial_{k}\left(R_{ij}v_{k}\right)-R_{kj}\partial_{i}v_{k}-R_{ki}\partial_{j}v_{k}\\ & & -\frac{\Lambda_{R}}{Tm}\left(R-k_{B}T\rho I\right), \end{array}$$
from which the tendency to relaxation of the Reynolds stress tensor to the respective MaxEnt value is obvious.

#### 4.2.2. DynMaxEnt to Hydrodynamics

As in the case of the conformation tensor in Section 4.1, let us now show how the Reynolds stress relaxes. The MaxEnt value (Equation 49) can be plugged into Equations (53). Assuming again isothermal incompressible flow, ρ=const, $\nabla \cdot u^{†}=0$ and $S_{e}=T=\mathrm{const}$, the equations become
(55a)$$\begin{array}{ccc} \frac{\partial u_{i}}{\partial t} & = & -\partial_{j}\left(u_{i}u_{j}^{†}\right)-\rho \partial_{i}\rho^{†}-u_{j}\partial_{i}u_{j}^{†}-s\partial_{i}s^{†}-\\ & & -k_{B}T\rho \partial_{i}R_{kk}^{†}-k_{B}T\rho \partial_{k}\left(R_{ki}^{†}+R_{ik}^{†}\right), \end{array}$$
(55b)$$\begin{array}{ccc} 0 & = & -k_{B}T\rho \left(\partial_{i}u_{j}^{†}+\partial_{j}u_{i}^{†}\right)-2\Lambda_{R}k_{B}\rho R_{ji}^{†}. \end{array}$$

The last equation has a solution
(56)$${\tilde{R}}^{†}=-\frac{T}{2\Lambda_{R}}\left(\nabla u^{†}+{\left(\nabla u^{†}\right)}^{T}\right)\;\mathrm{and}\;\mathrm{Tr}\tilde{R}=0.$$

Plugging this solution back into the equation for u leads to
(57)$$\begin{array}{ccc} \frac{\partial u_{i}}{\partial t} & = & -\partial_{j}\left(u_{i}u_{j}^{†}\right)-\rho \partial_{i}\rho^{†}-u_{j}\partial_{i}u_{j}^{†}-s\partial_{i}s^{†}\\ & & +\frac{k_{B}T^{2}\rho }{\Lambda_{R}}\partial_{k}\left(\partial_{k}u_{i}^{†}+\partial_{i}u_{k}^{†}\right), \end{array}$$
which is again the Navier–Stokes equation with shear viscosity. Relaxation of Reynolds stress thus leads to extra (also called turbulent) viscosity by means of the Dynamic MaxEnt reduction.

### 4.3. Hyperbolic Heat Conduction

#### 4.3.1. Non-Equilibrium Thermodynamics of Heat

Kinematics of heat transfer can be thought of as kinematics of phonons, and kinematics of phonons has been successfully described by Boltzmann-like dynamics, where the distribution function of phonons plays the role of state variable, see e.g., book [31]. By the reduction from the kinetic theory to fluid mechanics, see e.g., [1,14], kinematics of phonons can be expressed in terms of the entropy density and momentum related to entropy transport, the kinematics of which is expressed by a hydrodynamic-like Poisson bracket (see [32]). Subsequent transformation to density of matter ρ, total momentum of matter and phonons m, entropy density *s* and conjugate entropy flux w, which is equal to the ratio of phonon momentum and entropy density, leads to bracket
(58)$$\begin{array}{ccc} {\left\{F,G\right\}}^{\left(Cat\right)} & = & {\left\{F,G\right\}}^{\left(FM\right)}{|}_{u=m}+\int dr\left(G_{w_{i}}\partial_{i}F_{s}-F_{w_{i}}\partial_{i}G_{s}\right)\\ & & +\int drw_{j}\left(\partial_{i}F_{w_{i}}G_{m_{j}}-\partial_{i}G_{w_{i}}F_{m_{j}}\right)\\ & & +\int dr\left(\partial_{i}w_{j}-\partial_{j}w_{i}\right)\left(F_{w_{i}}G_{m_{j}}-G_{w_{i}}F_{m_{j}}\right)\\ & & +\int dr\frac{1}{s}\left(\partial_{i}w_{j}-\partial_{j}w_{i}\right)F_{w_{i}}G_{w_{j}}, \end{array}$$
expressing kinematics of matter and heat—the Cattaneo Poisson bracket. The name Cattaneo is due to the implied hyperbolicity of heat transport [19]. The Poisson bracket (Equation 58) generates reversible evolution equations
(59a)$$\begin{array}{ccc} \frac{\partial \rho }{\partial t} & = & -\partial_{k}\left(\rho E_{m_{k}}\right), \end{array}$$
(59b)$$\begin{array}{ccc} \frac{\partial m_{i}}{\partial t} & = & -\partial_{j}\left(m_{i}E_{m_{j}}\right)-\partial_{j}\left(w_{i}E_{w_{j}}\right)-\rho \partial_{i}E_{\rho }-m_{j}\partial_{i}E_{m_{j}}-s\partial_{i}E_{s}\\ & & -w_{k}\partial_{i}E_{w_{k}}+\partial_{i}\left(E_{w_{k}}w_{k}\right), \end{array}$$
(59c)$$\begin{array}{ccc} \frac{\partial s}{\partial t} & = & -\partial_{k}\left(sE_{m_{k}}+E_{w_{k}}\right), \end{array}$$
(59d)$$\begin{array}{ccc} \frac{\partial w_{k}}{\partial t} & = & -\partial_{k}E_{s}-\partial_{k}\left(w_{j}E_{m_{j}}\right)+\left(\partial_{k}w_{j}-\partial_{j}w_{k}\right)\left(E_{m_{j}}+\frac{1}{s}E_{w_{j}}\right). \end{array}$$

These evolution equations express reversible dynamics of fluid mechanics and conjugate entropy flux w. Note that, for the heat flux, i.e., flux of energy, the usual relation $q=E_{s}E_{w_{k}}=TJ^{\left(s\right)}$ holds true, see [14] for more details.

Local dissipation is enforced by adopting an algebraic dissipation potential, the simplest of which is
(60)$$\Xi \left(w^{*}\right)=\int dr\;\frac{1}{2}\frac{1}{\tau }{\left(w^{*}\right)}^{2}=\int dr\;\frac{1}{2}\frac{1}{\tau }{\left(\frac{1}{s^{†}}w^{†}\right)}^{2}\;,$$
where the last equality follows from the transformation between energetic and entropic representation. Then, the irreversible terms generated by this dissipation potential are added to the reversible evolution, Equations (59),
(61a)$$\begin{array}{cc} \frac{\partial \rho }{\partial t} & =-\partial_{k}\left(\rho m_{k}^{†}\right), \end{array}$$
(61b)$$\begin{array}{c} \begin{array}{cc} \frac{\partial m_{i}}{\partial t} & =-\partial_{j}\left(m_{i}m_{j}^{†}\right)-\partial_{j}\left(w_{i}w_{j}^{†}\right)-\rho \partial_{i}\rho^{†}-m_{j}\partial_{i}m_{j}^{†}-s\partial_{i}s^{†}\\ & \;-w_{k}\partial_{i}w_{k}^{†}+\partial_{i}\left(w_{k}^{†}w_{k}\right), \end{array} \end{array}$$
(61c)$$\begin{array}{cc} \frac{\partial s}{\partial t} & =-\partial_{k}\left(sm_{k}^{†}+w_{k}^{†}\right)+\frac{1}{\tau {\left(s^{†}\right)}^{2}}{\left(w^{†}\right)}^{2}, \end{array}$$
(61d)$$\begin{array}{c} \begin{array}{cc} \frac{\partial w_{k}}{\partial t} & =-\partial_{k}s^{†}-\partial_{k}\left(w_{j}m_{j}^{†}\right)+\left(\partial_{k}w_{j}-\partial_{j}w_{k}\right)\left(m_{j}^{†}+\frac{1}{s}w_{j}^{†}\right)\\ & \;-\frac{1}{\tau s^{†}}w_{k}^{†}\;. \end{array} \end{array}$$

These are the GENERIC equations for fluid mechanics with hyperbolic heat conduction.

#### 4.3.2. DynMaxEnt to Fourier Heat Conduction

The simplest possible dependence of entropy on the extra field w is quadratic (keeping in mind that *S* has to be a concave and even with respect to time reversal functional),
(62)$$S\left(\rho ,m,e,w\right)=\int dr\;s\left(\rho ,e-\frac{m^{2}}{2\rho }-\frac{1}{2}\alpha w^{2}\right)\;.$$

Consequently, entropy is highest (for given fields ρ, u and *e*) at w=0, which is the MaxEnt estimate $\tilde{w}$. Plugging this value into Equations (61) leads to
(63a)$$\begin{array}{cc} \frac{\partial \rho }{\partial t} & =-\partial_{k}\left(\rho m_{k}^{†}\right), \end{array}$$
(63b)$$\begin{array}{cc} \frac{\partial m_{i}}{\partial t} & =-\partial_{j}\left(m_{i}m_{j}^{†}\right)-\rho \partial_{i}\rho^{†}-m_{j}\partial_{i}m_{j}^{†}-s\partial_{i}s^{†}, \end{array}$$
(63c)$$\begin{array}{cc} \frac{\partial s}{\partial t} & =-\partial_{k}\left(sm_{k}^{†}+w_{k}^{†}\right)+\frac{1}{\tau {\left(s^{†}\right)}^{2}}{\left(w^{†}\right)}^{2}, \end{array}$$
(63d)$$\begin{array}{cc} 0 & =-\partial_{k}s^{†}-\frac{1}{\tau s^{†}}w_{k}^{†}\;. \end{array}$$

Equation ([Disp-formula FD63d-entropy-21-00715]) has the solution
(64)$${\tilde{w}}^{†}=-\tau s^{†}\nabla s^{†}\;.$$

Plugging this value into the rest of Equations (63), we obtain
(65a)$$\begin{array}{cc} \frac{\partial \rho }{\partial t} & =-\partial_{i}\left(\rho E_{m_{i}}\right), \end{array}$$
(65b)$$\begin{array}{cc} \frac{\partial m_{i}}{\partial t} & =-\partial_{j}\left(m_{i}E_{m_{j}}\right)-\rho \partial_{i}E_{\rho }-m_{j}\partial_{i}E_{m_{j}}-s\partial_{i}E_{s}, \end{array}$$
(65c)$$\begin{array}{cc} \frac{\partial s}{\partial t} & =-\partial_{k}\left(sE_{m_{k}}-\tau E_{s}\partial_{k}E_{s}\right)+\tau {\left(\nabla E_{s}\right)}^{2}\;, \end{array}$$
which are Euler equations with Fourier heat conduction. Indeed, denoting local temperature $E_{s}$ as *T*, the entropy flux is
(66)$$J^{\left(s\right)}=-\tau T\nabla T=\frac{-\lambda \nabla T}{T}\;,$$
where $\lambda =T^{2}\tau$ is the heat conductivity and q=−λ∇T is the heat flux.

One can also seek higher order corrections by means of the infinite DynMaxEnt chain. The corrected value of w implied by Equation ([Disp-formula FD63d-entropy-21-00715]) is
(67)$${\tilde{w}}^{\left(2\right)}=-\frac{\tau }{\alpha }T\nabla T,$$
which leads to energy
(68)$$E=\int dr\left(\frac{m^{2}}{2\rho }+\varepsilon \left(\rho ,s\right)+\frac{1}{2}\tau \alpha \varepsilon_{s}^{2}{\left(\nabla \varepsilon_{s}\right)}^{2}\right).$$

This is a weakly non-local energy implied by the second-order DynMaxEnt correction.

In summary, Fourier’s law, which tells us that heat flows from a hotter body to a colder body, can be derived by the dynamic MaxEnt reduction from the coupled dynamics of phonons and fluid mechanics. The only irreversibility on the higher level of description is present in the evolution equation for w. After the reduction, this irreversibility leads to irreversible terms in the equation for entropy (irreversible entropy flux and entropy production). Note that this reduction is again compatible with the asymptotic expansion carried out in [33].

### 4.4. Magnetohydrodynamics

Let us now turn to DynMaxEnt when the electromagnetic field is present.

#### 4.4.1. Non-Equilibrium Thermodynamics of Charged Mixtures

Dynamics of charged mixtures is governed by the Maxwell equations interacting with fluid mechanics of the species, see ([14] Section 6.4). Let us start with a Poisson bracket for the binary mixture of oppositely charged species endowed with a single entropy, total momentum density, displacement field and magnetic field, that is,
(69)$$\begin{array}{cc} {\left\{F,G\right\}}^{\left(\mathrm{EMHD}-2\right)}(\rho_{+},\rho_{-},m,s,D, & {B)=\left\{F,G\right\}}^{\left(\mathrm{CIT}-2\right)}\left(\rho_{+},\rho_{-},m,s\right)\\ {\left\{F,G\right\}}^{\left(\mathrm{EM}\right)}\left(D,B\right)\{ & +\int dr\left(F_{D_{i}}\varepsilon_{ijk}\partial_{j}G_{B_{k}}-G_{D_{i}}\varepsilon_{ijk}\partial_{j}F_{B_{k}}\right)\\ {\left\{F,G\right\}}^{\left(\mathrm{SP}\right)}\left(D,m\right)\{ & \begin{array}{l} +\int drD_{i}\left(\partial_{j}F_{D_{i}}G_{m_{j}}-\partial_{j}G_{D_{i}}F_{m_{j}}\right)\\ +\int dr\partial_{j}D_{j}\left(F_{m_{i}}G_{D_{i}}-G_{m_{i}}F_{D_{i}}\right)\\ +\int drD_{j}\left(F_{m_{i}}\partial_{j}G_{D_{i}}-G_{m_{i}}\partial_{j}F_{D_{i}}\right) \end{array}\\ {\left\{F,G\right\}}^{\left(\mathrm{SP}\right)}\left(B,m\right)\{ & \begin{array}{l} +\int drB_{i}\left(\partial_{j}F_{B_{i}}G_{m_{j}}-\partial_{j}G_{B_{i}}F_{m_{j}}\right)\\ +\int dr\partial_{j}B_{j}\left(F_{m_{i}}G_{B_{i}}-G_{m_{i}}F_{B_{i}}\right)\\ +\int drB_{j}\left(F_{m_{i}}\partial_{j}G_{B_{i}}-G_{m_{i}}\partial_{j}F_{B_{i}}\right), \end{array} \end{array}$$
where the CIT2 (binary classical irreversible thermodynamics [1,34]) bracket stands for
(70)$$\begin{array}{ccc} {\left\{F,G\right\}}^{\left(\mathrm{CIT}-2\right)}\left(\rho_{+},\rho_{-},m,s\right) & = & \int dr\;\rho_{+}\left(\partial_{i}F_{\rho_{+}}G_{m_{i}}-\partial_{i}G_{\rho_{+}}F_{m_{i}}\right)\\ & & +\int dr\;\rho_{-}\left(\partial_{i}F_{\rho_{-}}G_{m_{i}}-\partial_{i}G_{\rho_{-}}F_{m_{i}}\right)\\ & & +\int dr\;m_{i}\left(\partial_{j}F_{m_{i}}G_{m_{j}}-\partial_{j}G_{m_{i}}F_{m_{j}}\right)\\ & & +\int dr\;s\left(\partial_{i}F_{s}G_{m_{i}}-\partial_{i}G_{s}F_{m_{i}}\right), \end{array}$$
cf. bracket (Equation A3).

This system is additionally required to satisfy the Gauß laws for electric and magnetic charge, respectively. We have
(71)$$\begin{array}{c} \partial_{i}D_{i}=qe_{0}\left(\frac{\rho_{+}}{m_{+}}-\frac{\rho_{-}}{m_{-}}\right)\;\mathrm{and}\;\partial_{i}B_{i}=0\;, \end{array}$$
where the right-hand side of Equation (Equation 71)_left_ is the free charge density.

Total density $\rho =\rho_{+}+\rho_{-}$ and the free charge density can be used for the description instead of $\rho_{+}$ and $\rho_{-}$. Such transformation allows for the projection to the state variables without free charge density (i.e., where free charge density is relaxed) by letting the functionals depend only on (ρ,m,s,D,B). Consequently, bracket (Equation 69) transforms into
(72)$$\begin{array}{cc} {\left\{F,G\right\}}^{\left(\mathrm{EMHD}\right)}(\rho ,m,s,D, & {B)=\left\{F,G\right\}}^{\left(\mathrm{FM}\right)}\left(\rho ,m,s\right)+{\left\{F,G\right\}}^{\left(\mathrm{EM}\right)}\left(D,B\right)\\ & +{\left\{F,G\right\}}^{\left(\mathrm{SP}\right)}\left(D,m\right)+{\left\{F,G\right\}}^{\left(\mathrm{SP}\right)}\left(B,m\right)\;, \end{array}$$
and it is equipped with the updated constraint on the displacement field (given by the relaxed value of free charge density, typically zero) and magnetic field. Bracket (Equation 72) describes reversible evolution of electroneutral continuum coupled with the Maxwell equations.

Although the continuum described by bracket (Equation 72) is electroneutral, it can conduct electric current. This can be seen as a dissipation of the displacement field as suggested in [35]. Let us define dissipation potential
(73)$$\begin{array}{c} \Xi \left(D^{*}\right)=\int dr\frac{\sigma }{2}{\left(D^{*}\right)}^{2}=\int dr\frac{\sigma }{2}{\left(\frac{D^{†}}{s^{†}}\right)}^{2}\;. \end{array}$$

The reversible evolution generated by (Equation 72) and the irreversible evolution due to (Equation 73) give together
(74a)$$\begin{array}{cc} \frac{\partial \rho }{\partial t} & =-\partial_{k}\left(\rho m_{k}^{†}\right), \end{array}$$
(74b)$$\begin{array}{c} \begin{array}{cc} \frac{\partial m_{i}}{\partial t} & =-\partial_{j}\left(m_{i}m_{j}^{†}\right)-\rho \partial_{i}\rho^{†}-m_{j}\partial_{i}m_{j}^{†}-s\partial_{i}s^{†}-D_{j}\partial_{i}D_{j}^{†}-B_{j}\partial_{i}B_{j}^{†}\\ & \;+\partial_{j}\left(D_{j}D_{i}^{†}+B_{j}B_{i}^{†}\right), \end{array} \end{array}$$
(74c)$$\begin{array}{cc} \frac{\partial s}{\partial t} & =-\partial_{k}\left(sm_{k}^{†}\right)+\frac{\sigma }{{\left(s^{†}\right)}^{2}}{\left(D^{†}\right)}^{2}, \end{array}$$
(74d)$$\begin{array}{cc} \frac{\partial D_{i}}{\partial t} & =\varepsilon_{ijk}\partial_{j}B_{k}^{†}-\partial_{j}\left(D_{i}m_{j}^{†}-m_{i}^{†}D_{j}\right)-m_{i}^{†}\partial_{j}D_{j}-\frac{\sigma }{s^{†}}D_{i}^{†}, \end{array}$$
(74e)$$\begin{array}{cc} \frac{\partial B_{i}}{\partial t} & =-\varepsilon_{ijk}\partial_{j}D_{k}^{†}-\partial_{j}\left(B_{i}m_{j}^{†}-m_{i}^{†}B_{j}\right)-m_{i}^{†}\partial_{j}B_{j}. \end{array}$$

The conjugate displacement field $D^{†}$ is interpreted as electric intensity E, and the irreversible (last) term in Equation ([Disp-formula FD74d-entropy-21-00715]) then tells that the electric current is $J=\frac{\sigma }{s^{†}}E$. Dissipation in the D field can be thus seen as Ohm’s law (Alternatively, Ohm’s law can be derived from the relaxation of matter in the presence of the electric field, see [14].).

#### 4.4.2. DynMaxEnt to the Displacement Field—Passage to MHD

Let us apply the DynMaxEnt reduction to the displacement field in Equations (Equation 73) so that we approach the magnetohydrodynamics (MHD). The reversible part of the MHD equations can be obtained easily by projection to variables (ρ,m,s,B) as in [14], but we wish to also obtain the irreversible part of the equations.

Assuming energy quadratic in D, the MaxEnt value is $\tilde{D}=0$, which erases all terms containing the displacement field (but not $D^{†}$) from the equations. Except for the evolution of entropy, $D^{†}$ only remains in Equation ([Disp-formula FD74d-entropy-21-00715]), which can be solved and gives
(75)$$\begin{array}{c} {\tilde{D_{i}}}^{†}=\frac{s^{†}}{\sigma }\varepsilon_{ijk}\partial_{j}B_{k}^{†}\;. \end{array}$$
Introducing the curl of Equation (Equation 75) into Equation ([Disp-formula FD74d-entropy-21-00715]) yields a dissipative evolution equation for the magnetic field. The complete set of equations after this reduction reads
(76a)$$\begin{array}{cc} \frac{\partial \rho }{\partial t} & =-\partial_{k}\left(\rho m_{k}^{†}\right), \end{array}$$
(76b)$$\begin{array}{cc} \frac{\partial m_{i}}{\partial t} & =-\partial_{j}\left(m_{i}m_{j}^{†}\right)-\rho \partial_{i}\rho^{†}-m_{j}\partial_{i}m_{j}^{†}-s\partial_{i}s^{†}-B_{j}\partial_{i}B_{j}^{†}+\partial_{j}\left(B_{j}B_{i}^{†}\right), \end{array}$$
(76c)$$\begin{array}{cc} \frac{\partial s}{\partial t} & =-\partial_{k}\left(sm_{k}^{†}\right)+\frac{1}{\sigma }{\left(\nabla \times B^{†}\right)}^{2}, \end{array}$$
(76d)$$\begin{array}{cc} \frac{\partial B_{i}}{\partial t} & =-\partial_{j}\left(B_{i}m_{j}^{†}-m_{i}^{†}B_{j}\right)-m_{i}^{†}\partial_{j}B_{j}-\varepsilon_{ijk}\partial_{j}\left(\frac{s^{†}}{\sigma }\varepsilon_{klm}\partial_{l}B_{m}^{†}\right), \end{array}$$
which is compatible with [36]. The first terms on the right-hand side of Equation ([Disp-formula FD76d-entropy-21-00715]) can be further simplified for constant $\sigma /s^{†}$, provided that the contribution of the magnetic field to the energy is $\frac{B^{2}}{\mu_{0}}$. We can then write
(77)$$\begin{array}{cc} \frac{\partial B}{\partial t} & =\frac{s^{†}}{\mu_{0}\sigma }\Delta B-\nabla \times \left(B\times m^{†}\right)\;, \end{array}$$
where $\partial_{i}B_{i}=0$ was finally used. Keeping in mind that $m^{†}=v$, Equation (Equation 77) is the advection–diffusion equation for the magnetic field, *cf.* ([37] Equation 2.15) or [38]. The coefficient $\frac{s^{†}}{\mu_{0}\sigma }$ is referred to as the magnetic diffusivity.

In summary, after the relaxation of free charge density, one gets the electroneutral continuum coupled with Maxwell equations. Further dissipation in the displacement field then leads by the DynMaxEnt reduction to the MHD equations including magnetic diffusivity.

## 5. Contact Geometry

Contact geometry seems to be the so far most general geometric formulation of non-equilibrium thermodynamics. It started with works of Hermann [11] in equilibrium thermodynamics and continued to non-equilibrium thermodynamics, e.g., [12,15,39,40,41] and many others. Here, we adopt a recent version of contact-geometric formulation of GENERIC from [14].

## 6. Contact GENERIC

We begin by introducing a space
(78)$$M^{\left(cont\right)}=M\times M^{*}\times N_{eq}^{*}\times N_{eq}\times R$$
with coordinates $\left(x,x^{*},y^{*},y,\phi \right)$. The space M with elements *x* is the state space, the space $M^{*}$ with elements $x^{*}$ is its dual. Similarly, $N_{eq}$ with elements *y* is the state space on the equilibrium level, $N_{eq}^{*}$ with elements $y^{*}$ is its dual. We recall that y=(E,N) and $y^{*}=\left(E^{*},N^{*}\right)$, where $E^{*}=\frac{1}{T}$ and $N^{*}=-\frac{\mu }{T}$. We moreover introduce the fundamental thermodynamic relation S=S(x),y=y(x) represented in $M\times N_{eq}\times R$ by the **Gibbs manifold**$M^{\left(G\right)}$ that is the image of the mapping
(79)x↪(x,y(x),S(x)).

Corresponding to the fundamental thermodynamic relation is the thermodynamic potential $\Phi \left(x,y^{*}\right)=-S\left(x\right)+\langle y^{*},y\left(x\right)\rangle$, where 〈,〉 denotes the inner product.

The Gibbs manifold $M^{\left(G\right)}$ can be now extended to the **Gibbs–Legendre manifold**$M^{\left(GL\right)}$ (in the shorthand notation *GL manifold*) that is the image of the mapping
(80)$$\left(x,y^{*}\right)↪\left(x,\Phi_{x}\left(x,y^{*}\right),y^{*},\Phi_{y^{*}}\left(x,y^{*}\right),\Phi \left(x,y^{*}\right)\right)$$
in $M^{\left(cont\right)}$.

The thermodynamics in M is completely expressed in the GL manifold $M^{\left(GL\right)}$. Note that ${\left[M^{\left(GL\right)}\right]}_{y^{*}=0}$ (i.e., the image of the mapping
(81)$$\left(x,0\right)↪(x,-S_{x}\left(x\right),0,y\left(x\right),-S\left(x\right))$$
in the space $M\times M^{*}\times N_{eq}\times R$) is an extension of the Gibbs manifold $M^{\left(G\right)}$ by including the conjugate variable $x^{*}$. Moreover, the manifold ${\left[M^{\left(GL\right)}\right]}_{x^{*}=S_{x}=0}$ displays the states $x_{eq}\left(y^{*}\right)$ that represent in M the equilibrium states and also the fundamental thermodynamic relation $S^{*}\left(y^{*}\right),y\left(y^{*}\right)$ in $N_{eq}$ implied by the fundamental thermodynamic relation S(x),y(x) in M. Indeed, ${\left[M^{\left(GL\right)}\right]}_{x^{*}=0}$ is the image of the mapping
(82)$$\left(x,y^{*}\right)↪\left(x_{eq}\left(y^{*}\right),0,y^{*},y\left(x_{eq}\left(y^{*}\right)\right),S^{*}\left(y^{*}\right)\right).$$

Let us turn to the time evolution in $M^{\left(cont\right)}$. We begin by introducing in $M^{\left(cont\right)}$ a bracket
(83)$$\begin{array}{ccc} {\left\{A,B\right\}}^{\left(cont\right)} & = & \left(\langle A_{x},B_{x^{*}}\rangle -\langle B_{x},A_{x^{*}}\rangle \right)\\ & & -(\langle A_{y},B_{y^{*}}\rangle -\langle B_{y},A_{y^{*}}\rangle )\\ & & -(\langle x^{*},A_{x^{*}}\rangle B_{\phi }-\langle x^{*},B_{x^{*}}\rangle A_{\phi })\\ & & +(AB_{\phi }-BA_{\phi })\\ & & +(\langle A_{y},y\rangle B_{\phi }-\langle B_{y},y\rangle A_{\phi }), \end{array}$$
where *A* and *B* are sufficiently regular functions $M^{\left(cont\right)}\to R$. This bracket consists of two contact brackets (95) of paper [41]. With such bracket, we introduce the time evolution in $M^{\left(cont\right)}$ by an equation
(84)$$\dot{A}={\left\{A,H^{\left(cont\right)}\right\}}^{\left(cont\right)}-AH_{\phi }^{\left(cont\right)}$$
that is required to hold for all *A*. The function $H^{\left(cont\right)}:M^{\left(cont\right)}\to R$, called a contact Hamiltonian, will be specified below. The last term on the right-hand side corresponds to the non-conservation of the phase-space volume [40]. Written explicitly, the time evolution Equation (Equation 84) take the form
(85a)$$\begin{array}{ccc} \dot{x} & = & H_{x^{*}}^{\left(cont\right)}, \end{array}$$
(85b)$$\begin{array}{ccc} \dot{x^{*}} & = & -H_{x}^{\left(cont\right)}-x^{*}H_{\phi }^{\left(cont\right)}, \end{array}$$
(85c)$$\begin{array}{ccc} \dot{y^{*}} & = & H_{y}^{\left(cont\right)}, \end{array}$$
(85d)$$\begin{array}{ccc} \dot{y} & = & -H_{y^{*}}^{\left(cont\right)}+yH_{\phi }^{\left(cont\right)}, \end{array}$$
(85e)$$\begin{array}{ccc} \dot{\phi } & = & -H^{\left(cont\right)}+\langle x^{*},H_{x^{*}}^{\left(cont\right)}\rangle -\langle H_{y}^{\left(cont\right)},y\rangle . \end{array}$$

These are the evolution equations in M.

Next, we specify the contact Hamiltonian $H^{\left(cont\right)}$
(86)$$H^{\left(cont\right)}\left(x,x^{*},y^{*},y,\phi \right)=-S^{\left(cont\right)}\left(x,x^{*},y^{*}\right)+\frac{1}{E^{*}}E^{\left(cont\right)}\left(x,x^{*},y^{*}\right),$$
where
(87)$$\begin{array}{ccc} S^{\left(cont\right)}\left(x,x^{*},y^{*}\right) & = & \Xi \left(x,x^{*},y^{*}\right)-{\left[\Xi \left(x,x^{*},y^{*}\right)\right]}_{x^{*}=\Phi_{x}},\\ E^{\left(cont\right)}\left(x,x^{*},y^{*}\right) & = & \langle x^{*},L\Phi_{x}\rangle . \end{array}$$

Ξ is the dissipation potential entering GENERIC and *L* is the Poisson bivector also entering GENERIC. Both Ξ and *L* are degenerate in the sense
(88)$$\begin{array}{c} \langle x^{*},LS_{x}\rangle =\langle x^{*},LN_{x}\rangle =0,\;\;\;\;\forall x^{*},\\ \langle E_{x},\Xi_{x^{*}}\rangle =\langle N_{x},\Xi_{x^{*}}\rangle =0,\;\;\;\;\forall x^{*},\\ \langle x^{*},{\left[\Xi_{x^{*}}\right]}_{x^{*}=E_{x}}\rangle =\langle x^{*},{\left[\Xi_{x^{*}}\right]}_{x^{*}=M_{x}}\rangle =0,\;\;\;\;\forall x^{*}. \end{array}$$

We note in particular that the contact Hamiltonian (Equation 86) is independent of *y* and ϕ.

With (Equation 86), the time evolution Equations ([Disp-formula FD85a-entropy-21-00715]) become
(89a)$$\begin{array}{ccc} \dot{x} & = & \frac{1}{E^{*}}L\Phi_{x}-\Xi_{x^{*}}, \end{array}$$
(89b)$$\begin{array}{ccc} \dot{x^{*}} & = & \Phi_{xx}\left(\frac{1}{E^{*}}Lx^{*}-{\left[\Xi_{x^{*}}\right]}_{x^{*}=\Phi_{x}}\right)\\ & & -\frac{1}{E^{*}}\langle x^{*},L_{x}\Phi_{x}\rangle +\Xi_{x}-{\left[\Xi_{x}\right]}_{x^{*}=\Phi_{x}}, \end{array}$$
(89c)$$\begin{array}{ccc} \dot{\phi } & = & -\langle x^{*},\Xi_{x^{*}}+\Xi -{\left[\Xi \right]}_{x^{*}=\Phi_{x}}, \end{array}$$
(89d)$$\begin{array}{ccc} \dot{y^{*}} & = & 0, \end{array}$$
(89e)$$\begin{array}{ccc} \dot{y} & = & \Xi_{y^{*}}-\Xi_{y^{*}}{|}_{x^{*}=\Phi_{x}}. \end{array}$$

If we now evaluate ([Disp-formula FD89a-entropy-21-00715]) on the GL manifold $M^{\left(GL\right)}$ (note that ${\left[H^{\left(cont\right)}\right]}_{M^{\left(GL\right)}}=0$), we arrive at
(90a)$$\begin{array}{ccc} \dot{x} & = & \frac{1}{E^{*}}L\Phi_{x}-\Xi_{x^{*}}{|}_{\left(x^{*}=\Phi_{x},y=\Phi_{y^{*}}\right)}, \end{array}$$
(90b)$$\begin{array}{ccc} \dot{x^{*}} & = & \Phi_{xx}\left(\frac{1}{E^{*}}L\Phi_{x}-{\left[\Xi_{x^{*}}\right]}_{\left(x^{*}=\Phi_{x},y=\Phi_{y^{*}}\right)}\right), \end{array}$$
(90c)$$\begin{array}{ccc} \dot{\phi } & = & -\langle x^{*},\Xi_{x^{*}}\rangle {|}_{\left(x^{*}=\Phi_{x},y=\Phi_{y^{*}}\right)}, \end{array}$$
(90d)$$\begin{array}{ccc} \dot{y^{*}} & = & 0, \end{array}$$
(90e)$$\begin{array}{ccc} \dot{y} & = & 0, \end{array}$$
which are the GENERIC evolution equations. See [14] for more details.

### 6.1. Relation to DynMaxEnt

Consider state variables x=(r,p) as in the Section 3.3 on the damped particle (disregarding entropy for simplicity). The Poisson bivector is then canonical, $L=\left(\begin{array}{cc} 0 & 1\\ -1 & 0 \end{array}\right)$, and the dissipation potential is quadratic in conjugate momentum, $\Xi =\frac{1}{2\tau }{\left(p^{*}\right)}^{2}$. This conjugate momentum, according to Equation (89b), approaches the value where GENERIC evolution equations hold, Equation (11), and eventually reaches the value given by the derivative of thermodynamic potential.

Now, assume that state variable p evolves faster to the corresponding equilibrium (zero) than both r and conjugate variables. The GENERIC evolution equations then become Equations ([Disp-formula FD11a-entropy-21-00715]) and (Equation 15). The conjugate variable $p^{*}$ approaches the value where the GENERIC equations are valid, which means that it approaches solutions to Equation (Equation 15) as in the DynMaxEnt procedure. Conjugate variables approach the Gibbs–Legendre manifold (Equation 80). Contact geometry provides motivation and geometric meaning to the Dynamic MaxEnt reduction.

## 7. Conclusions

In this paper, we have presented a method for reduction of detailed dynamics to less detailed dynamics called Dynamic MaxEnt. The key feature of the method is that conjugate variables are promoted to independent variables and as such they can relax to a quasi-equilibrium in a different way than state variables. While relaxation of the state variables generates the entropy on the lower level of description, relaxation of conjugate variables ensures that the vector field on the higher level becomes tangent to the quasi-equilibrium manifold.

First, in Section 2, the usual MaxEnt is recalled, which gives state variables on the higher (detailed) level as functions of state variables on the lower (less detailed) level. The DynMaxEnt method is then presented in Section 3 including the infinite chain of higher order DynMaxEnt corrections, and it is compared to asymptotic expansions in Section 3.3.3. Then, the method is used on the reduction of dynamics of complex fluids equipped with conformation tensor and Reynolds stress to the Navier–Stokes equations, reduction of hyperbolic heat conduction to the Fourier law, where we again compare the result to the formal asymptotic methods, and reduction of electromagnetohydrodynamics to magnetohydrodynamics. Finally, motivation for the DynMaxEnt method by contact geometry is shown in Section 5.

In summary, this paper contains a relatively straightforward method for reduction from dynamics on a detailed level of description to dynamics on a less detailed level of description. 

## Figures and Tables

**Figure 1 entropy-21-00715-f001:**
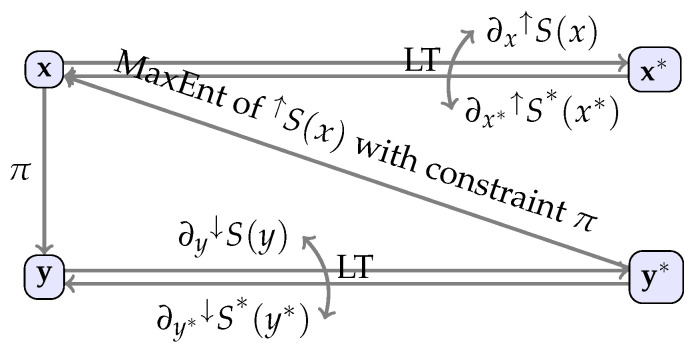
A summary of static MaxEnt highlighting relations between state variables on the higher level and the lower level of description and their conjugates. MaxEnt provides lower entropy ${}^{↓}S\left(y\right)$ and a relation $x=\pi^{-1}\left(y\right)$ from composition of $\tilde{x}\left(y^{*}\right)$ and $\tilde{y^{*}}\left(y\right)$. LT denotes a relation via Legendre transformation, π stands for a projection from the microscale to the macroscale variables and by an arrow we depict a mapping (written above or below the arrow) that relates the variables in the connected nodes.

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
