# Peer review of "Dynamic Maximum Entropy Reduction"

_entropy, 2019, doi:10.3390/e21070715_

Round 1

Reviewer 1 Report

The present manuscript is, maybe, too mathematical for my taste. But, the authors are well-known in the field and I found the subject matter adequate to the "entropy" readers. The paper language is clear and it reads well in spite of the very abstract nature of the topic. I found pleasing the effort to connect with applications in Section 4. I can only recommend the manuscript for publication in "entropy".

My only (non-specialist) concern is about the somewhat obscure issue of considering two velocities as independent, to only make them equal at the end of the day. As far as I know, Galilean invariance requires that only the center of mass velocity appears in the mass balance. Hence, the proposal of the authors may lead some (uninformed) readers to question the Galilean invariance of the Poisson bracket used as starting point of the developments presented in the manuscript. If the authors want to introduce a comment about this issue I would appreciate. But, of course, is not mandatory

Author Response

Thank you for pointing out the issue of transformation rules. A note has been added to the summary of DynMaxEnt. 

Reviewer 2 Report

The paper discusses a method called "DynMaxEnt", which is a coarse graining procedure mapping a microscopic description of a given dynamical system to a macroscopic one. The macroscopic description contains fewer degrees of freedom and the coarse graining is performed by maximizing the entropy with respect to the degrees of freedom that are "integrated out". As discussed in sec. 3, the method involves an infinite sequence of recursion relations. The rest of sec. 3 as well as sec. 4  implement the method through various examples and discuss relations with other approaches. 

The structure of the article is clear, the results are quite well presented and are sufficiently useful for further research on the topic. I therefore recommend the paper for publication, after the following comments have been addressed:

1. It would be good to include an outline of the paper at the end of the introduction.

2. In eq. (14), first matrix on the left-hand side: the last row should probably be (0,1) rather than (1,0). In the same equation, the last vector on the left-hand side should have s^+ in the denominator of the second row, not r^+.

3. In eq. (21), the right-hand side seems to contain several typos, in particular the second term should be proportional to \zeta^3 and the third one to \zeta^5.

Author Response

Thank you for reviewing our manuscript. We have included the outline in the new version.

We have also corrected Eqs. 14 and 21. Thank you for pointing out those typos.